# Effect of rLH Supplementation during Controlled Ovarian Stimulation for IVF: Evidence from a Retrospective Analysis of 1470 Poor/Suboptimal/Normal Responders Receiving Either rFSH plus rLH or rFSH Alone

**DOI:** 10.3390/jcm11061575

**Published:** 2022-03-13

**Authors:** Stefano Canosa, Andrea Roberto Carosso, Noemi Mercaldo, Alessandro Ruffa, Francesca Evangelista, Francesca Bongioanni, Chiara Benedetto, Alberto Revelli, Gianluca Gennarelli

**Affiliations:** 1Obstetrics and Gynecology 1U, Physiopathology of Reproduction and IVF Unit, Department of Surgical Sciences, Sant’Anna Hospital, University of Turin, 10126 Turin, Italy; andrea88.carosso@gmail.com (A.R.C.); noemimercaldo@gmail.com (N.M.); alessandro.ruffa92@gmail.com (A.R.); francesca-evangelista@hotmail.it (F.E.); chiara.benedetto@unito.it (C.B.); gennarelligl@gmail.com (G.G.); 2Livet, GeneraLife IVF, 10126 Turin, Italy; francesca.bongio@gmail.com; 3Obstetrics and Gynecology 2U, Department of Surgical Sciences, Sant’Anna Hospital, University of Turin, 10126 Turin, Italy; aerre99@yahoo.com

**Keywords:** recombinant FSH, recombinant LH, in vitro fertilization, IVF outcome, live birth rate, poor responders, suboptimal responders, Bologna criteria, POSEIDON classification

## Abstract

We retrospectively studied a real-life population of 1470 women undergoing IVF, with poor/suboptimal/normal ovarian responsiveness to controlled ovarian stimulation (COS), comparing the cumulative live birth rate (cLBR) when COS was performed using rFSH alone or rFSH + rLH in a 2:1 ratio. Overall, we observed significantly higher cLBR in the rFSH alone group than in the rFSH + rLH group (29.3% vs. 22.2%, *p* < 0.01). However, considering only suboptimal/poor responders (*n* = 309), we observed comparable cLBR (15.6% vs. 15.2%, *p* = 0.95) despite the fact that patients receiving rFSH + rLH had significantly higher ages and worse ovarian reserve markers. The equivalent effectiveness of rFSH + rLH and rFSH alone was further confirmed after stratification according to the number of oocytes retrieved: despite basal characteristics were still in favor of rFSH alone group, the cLBR always resulted comparable. Even subdividing patients according to the POSEIDON classification, irrespective of differences in the baseline clinical characteristics in favor of FSH alone group, the cLBR resulted comparable in all subgroups. Despite the retrospective, real-life analysis, our data suggest that rLH supplementation in COS may represent a reasonable option for patients with predictable or unexpected poor/suboptimal ovarian responsiveness to FSH, those matching the Bologna criteria for poor responsiveness, and those included in the POSEIDON classification.

## 1. Introduction

Female age represents the most critical factor affecting both natural fecundity and pregnancy chance in in vitro fertilization (IVF) [1,2]. Indeed, the relevant and progressive decrease of female fertility observed in advanced reproductive ages (>35 years) is mainly due to age-related detrimental effects on oocyte quality, which ultimately affects embryo competence [3]. A higher incidence of chromosomal abnormalities in oocytes [4], together with a progressive decline in reproductive endocrine function [5], appear to be the main driving forces of fertility impairment in these patients. Based on the “two-cell, two-gonadotropin” theory, follicle stimulating hormone (FSH) and luteinizing hormone (LH) play a critical role in stimulating the two cellular components of the ovary, granulosa cells and theca cells, respectively, leading to the secretion of ovarian steroids. FSH controls the proliferation of granulosa cells and promotes estradiol (E2) synthesis, whereas LH stimulates androgen production by theca cells [6,7]. Although FSH can induce follicular growth even without LH, it has been observed that, in the absence of LH activity, the follicles undergo suboptimal development [8], not only due to the shortage of androgen substrate for aromatization, but also because of the lack of a direct LH effect [9]. Indeed, LH plays pivotal roles in: (i) follicular recruitment, increasing FSH receptor expression in granulosa cells [10]; (ii) follicular maturation, via local growth factors recruitment [11]; (iii) completion of meiosis and extrusion of the first polar body [12]; and (iv) decidualization of endometrial stromal cells promoting embryo implantation [13]. Notably, both direct LH effects and androgen production seem to be perturbed by ovarian ageing [6]. In most women undergoing controlled ovarian stimulation (COS) for medically assisted reproduction (MAR), endogenous LH sustains follicle recruitment, but women above 35 years exhibit decreased LH activity, reflecting into lower androgen and estrogen follicular fluid levels [14]. In this scenario, human recombinant LH (rLH) supplementation during COS has been suggested as more effective than high-dose rFSH alone in improving clinical pregnancy rates in women of advanced reproductive age [15,16]. However, the role of rFSH + rLH co-treatment in these women is still a matter of debate, as no consensus on the indication of LH supplementation has been reached so far [17,18,19,20]. The aim of this retrospective study was to compare the live birth rate (LBR) of expected poor, suboptimal, and normal responders (classified according to the Bologna criteria and POSEIDON classification) when COS was performed using rFSH + rLH co-treatment or rFSH alone. The respective effectiveness of the two regimens was further tested after patients’ stratification for the number of retrieved oocytes.

## 2. Materials and Methods

### 2.1. Study Design

This large retrospective study included 1470 women aged 20–43 years (mean age 36.7 ± 4.1), with normal body mass index (BMI 18–25), and ovarian reserve markers suggesting a poor, suboptimal, or normal responsiveness to COS (AMH ≤ 2.5 ng/mL, AFC ≤ 15) [21,22]. Among a total 1612 women undergoing IVF in the time period of the study, 242 were excluded because not matching the inclusion criteria: patients with AMH and AFC above the mentioned limits, BMI outside the indicated limits, chronic anovulation, multifollicular or PCO ovary, as well as those with a history of severe ovarian hyperstimulation syndrome. All selected patients underwent IVF with their own oocytes at S. Anna Hospital IVF Unit (*n* = 1388) or at the associated Livet (GeneraLife IVF) private clinic (*n* = 382) between January 2004 and December 2019. According to the Bologna criteria [23] and to the POSEIDON classification [24], patients were classified as follows: “poor responders” (AFC ≤ 6, AMH ≤ 1.2 ng/mL, ≤3 retrieved oocytes), “suboptimal responders” (AFC ≤ 6, AMH ≤ 1.2 ng/mL, ≤6 oocytes) and “normal responders” (AFC 6–15, AMH 1.2–2.5 ng/mL, 6–15 oocytes). Expected normal responders showing a suboptimal ovarian response in a previous IVF cycle at standard stimulation dose were also considered as suboptimal responders. These patients were scheduled to undergo a new COS with either FSH alone (*n* = 669, rFSH alone Group) or rLH in addition to rFSH (*n* = 801, rFSH + rLH Group). Being a retrospective study based on the real life, daily clinical practice, rFSH+rLH Group was mainly composed by expected or proven poor/suboptimal responders, whereas rFSH alone Group included mainly expected normal responders. The patients’ clinical characteristics and the outcome of COS were recorded, including the total dose of exogenous FSH, peak estradiol (E2) levels, number of retrieved oocytes, ovarian sensitivity index (OSI = retrieved oocytes × 1000/total gonadotropin dose) [22] and fertilization rate. The cumulative live birth rate per oocyte retrieval (cLBR/OPU) was chosen as the primary outcome; secondary outcomes were: number of retrieved oocytes, number of mature oocytes, and proportion of top-scored embryos.

### 2.2. COS Regimen

COS was performed as previously described [21]. As a common background, the choice of the starting dose was based on age, body mass index (BMI), AFC, circulating AMH, as well as on the response to previous COS cycles. Patients in rFSH+rLH group (*n* = 801) received a subcutaneous starting dose of 150–300 IU/d rFSH + rLH 2:1 (Pergoveris^®^, Merck, Darmstadt, Germany), whereas patients in rFSH alone group (*n* = 669) received 150–300 IU/d recombinant FSH (rFSH; Gonal F^®^; Merck, Darmstadt, Germany). Both medications were administered within a “long” protocol with GnRH-agonists (408/669, 61% of patients in rFSH alone group; 529/801, 66% of patients in rFSH-rLH group) or a “short” protocol with GnRH-antagonists (261/669, 39% of patients in rFSH alone group; 272/801, 34% of patients in rFSH-rLH group). The classical “long” protocol was performed administering the GnRH-agonist buserelin (Suprefact^®^, Hoechst, Germany; 900 mcg/d intranasally) from day 21 of the incoming cycle. After approximately two weeks, pituitary suppression was verified (appearance of a menstrual bleeding, serum estradiol <50 pg/mL, endometrial thickness <3 mm) before starting COS. In the “short” protocol, the GnRH-antagonist cetrorelix (Cetrotide^®^, Merck-Serono, Germany) was started at a subcutaneous dose of 0.25 mg/d according to a flexible schedule, when at least one follicle ≥14 mm diameter was observed at ultrasound (US). Circulating E2 and transvaginal US examination were performed every second day from stimulation day 6–7 to monitor follicular growth, adapting the medication dose when required. When at least two follicles reached 18 mm mean diameter, with appropriate E2 levels, a single s.c. injection of 10,000 IU hCG (Gonasi HP, IBSA, Lugano, Switzerland) was administered in order to trigger ovulation.

### 2.3. Oocyte Retrieval, Fertilization, Embryo Culture and Transfer

Transvaginal ultrasound-guided oocyte pick-up (TV-US OPU) was performed 35–37 h after ovulation trigger, under local anesthesia (paracervical block). Follicular fluids were aspirated and immediately observed under a stereomicroscope. Cumulus-oocyte complexes (COCs) were washed in buffered medium (Flushing medium, Cook Ltd., Ireland), and within 4 h from OPU oocytes were inseminated using conventional IVF or ICSI according to the quality of the semen sample. Normal fertilization was confirmed when the presence of two pronuclei (2PN) and the extrusion of the second polar body were observed 16–18 h after oocyte insemination or microinjection. Fertilized eggs were cultured in pre-equilibrated Cleavage medium (Cook, Ireland) overlaid with mineral oil (Culture Oil, Cook Ltd., Limerick, Ireland) up to day 3 of development; at this stage, a change of medium was performed using a stage-specific medium (Blastocyst medium, Cook, Ireland) until the blastocyst stage. As usual practice in our lab, embryo morphological assessment was performed on day 2 using the Integrated Morphology Cleavage Score (IMCS) [25], and again on day 5 according to the Istanbul Consensus Workshop [26]. Embryo transfer (ET), in utero, was performed either with two top-quality embryos on day 3 or with a single good quality blastocyst on day 5; the choice was driven by the number of fertilized oocytes (<5 or ≥5, respectively), and of top-quality embryos on day 2 (<3 or ≥3, respectively). ET was performed using the Sydney Guardia soft catheter (Cook, Australia), under transvaginal US guidance. If several good scoring embryos were obtained, surplus embryos were frozen at the blastocyst stage on day 5 or 6, and kept in liquid nitrogen for further use. The luteal phase was supported administering 180 mg/d natural progesterone (Crinone 8^®^, Merck, Darmstadt, Germany) for 15 days. Pregnancy was assessed by serum hCG measurement after 15 days from ET, and then confirmed if at least one gestational sac was visualized at TV-US after two further weeks. Only cases with US confirmation were counted in the calculation of pregnancy rate, whereas biochemical pregnancies were not considered.

### 2.4. Statistical Analysis

The primary endpoint was the cumulative live birth rate per oocyte pick-up (cLBR/OPU); a real-life population of normal/suboptimal/poor responders was studied, and outcomes were analysed and compared according to the type of gonadotropin used in COS (rFSH + rLH vs. rFSH alone). The analysis was conducted first considering all patients, then considering only patients with 6 or less retrieved oocytes (poor/suboptimal responders). In addition, a further sub-analysis was performed, in which the two different COS regimens were compared after stratification for the number of retrieved oocytes or according to the POSEIDON subgroup. Due to the normal distribution of data in the Shapiro–Wilk test, continuous variables were expressed as mean ± standard deviation (SD), whereas categorical variables were expressed as absolute values and percentage. The comparison among groups was performed using the GraphPad Prism V7 software, applying the Student’s t parametric test, the non-parametric Mann–Whitney test, or the Chi-square test, as appropriate. All statistical tests were two-sided, and a *p* value ≤ 0.05 was considered statistically significant.

## 3. Results

The baseline clinical characteristics and IVF outcome of the whole patients’ group are shown in Table 1. As expected, due to the retrospective nature of the study, women receiving rFSH alone had significantly lower age and infertility duration, and significantly higher day 3 AMH and AFC, confirming how in the current clinical practice rLH supplementation is often reserved to patients with poorer prognosis indexes. According to these unfavorable prognostic features, the number of retrieved oocytes was significantly lower in rFSH + rLH group (*p* < 0.0001), as was the number of available metaphase II oocytes (*p* < 0.0001). Overall, fertilization and cleavage rates, the mean embryo morphological score and the proportion of top-scored embryos were comparable in the two groups, suggesting a comparable oocyte quality, but the higher availability of MII oocytes led to a significantly lower number of frozen embryos in the rFSH + rLH group (0.3 ± 0.9 vs. 0.6 ± 1.1, *p* < 0.0001). Finally, considering all patients we observed significantly lower cLBR/OPU in the rFSH + rLH group than in the rFSH alone group (22.2% vs. 29.3%, *p* < 0.01) (Table 1).

Excluding the analysis the normal responders (Table 2), and thus including only the 309 suboptimal/poor responders (263 of which received rFSH + rLH and 46 received rFSH alone), we observed that, in spite of a significantly higher mean age (38.3 ± 3.5 years vs 36.4 ± 4.3, *p* < 0.01), patients who received rFSH + rLH obtained a comparable number of oocytes and mature oocytes, produced an equivalent proportion of top-scored embryos, had a similar number of frozen embryos, and finally obtained a practically identical cLBR/OPU (15.6% vs. 15.2%, *p* = 0.95) than women receiving rFSH alone.

The equivalence between rFSH + rLH and rFSH alone was further confirmed after stratifying patients for the number of retrieved oocytes: although, in the resulting smaller subgroups, the differences in basal characteristics (in favor of rFSH alone group) were maintained, the cLBR/OPU resulted comparable (Table 3 and Figure 1).

Even subgrouping patients according to the POSEIDON classification, irrespective of differences in the baseline clinical characteristics and in the number of retrieved oocytes, favorable to rFSH alone group in all POSEIDON groups, the cLBR/OPU always resulted comparable (Table 4).

## 4. Discussion

The poor ovarian response to COS represents a major challenge in MAR, especially in women of advanced reproductive age, who also have progressively worse oocyte competence. The term “suboptimal response” is currently used to describe a condition of reduced sensitivity of the ovary to exogenous rFSH [27]; in concrete, it appears as a poor ovarian response to COS by women who require a higher-than-expected dose of rFSH and/or need to receive rLH supplementation [18]. In this study, we retrospectively analyzed the IVF outcome of a real-life, large population of poor/suboptimal/normal responders receiving COS with rFSH plus rLH or rFSH alone. Considering the whole group of enrolled patients, we observed a significantly reduced oocyte yield, number of mature eggs, and cLBR/OPU in women receiving rFSH plus rLH co-treatment, according to the presence of worse prognosis predictors in these patients (higher age, lower AMH and AFC) and to the widespread habit, in clinical daily practice, of supplementing rFSH with rLH more frequently in case of older women with poor prognosis markers. However, when only the suboptimal and poor responders (according to Bologna criteria and POSEIDON classification) were considered, we found that patients who received rLH supplementation, despite being significantly older, obtained a comparable oocyte yield, produced a similar proportion of mature eggs and top-scored embryos, had a similar number of frozen embryos, and finally obtained the same cLBR/OPU than their counterparts who received rFSH only. This finding agrees with a recently published systematic review and meta-analysis, showing that in women with advanced maternal age rFSH plus rLH co-treatment obtained similar clinical outcome [17]. These data suggest that LH supplementation may be an interesting option, particularly in the case of older patients with a poor or suboptimal response to FSH. The mechanism by which rLH might exert a beneficial effect in suboptimal and poor responders of more advanced reproductive age is not fully understood. Exogenous LH could increase thecal androgen production, which is known to decrease in advanced age [6], restoring the ideal follicular milieu within the developing follicles [28,29]. In addition, LH could counteract the age-related increase in the apoptosis rate of cumulus cells [30], promoting cell proliferation and oocyte cytoplasmic maturation [31]. Additionally, LH could act via up-regulation of some mediators involved in the pro-angiogenetic activity of follicular cells [32]. Taken together, these observations suggest that LH activity could act improving oocyte quality and competence, rather than oocyte yield. Furthermore, some studies suggest that endometrial maturation may be disturbed in case of LH deficiency [33], and exogenous LH could modulate signaling molecules involved in embryo-endometrium crosstalk [34,35]: this effect could explain the higher implantation rate observed in women of an advanced age supplemented with exogenous LH [20]. Finally, the premature progesterone (P) rise, responsible for the asynchrony of embryo and endometrial development leading to implantation impairment [36,37,38] is related to a high exogenous rFSH dose, and is counteracted by LH addition [39]. As the cLBR/OPU is known to increase with the number of retrieved oocytes [40], and having more available oocytes means increasing the chance of getting an euploid blastocyst [24], we performed a further sub-analysis in which the two COS regimens were compared considering patients stratified in small subgroups according to the number of available oocytes. Again, the cLBR/OPU resulted comparable in all subgroup pairs with identical number of oocytes, despite the higher age in the rFSH + rLH group, further suggesting that rLH supplementation in patients of advanced reproductive age could be linked to an improved oocyte quality, rather than to an improved oocyte yield. The rather recent POSEIDON classification, based on a combination of age and ovarian reserve markers, divides patients with poor ovarian responsiveness to FSH into two main categories, namely the “unexpected poor / suboptimal responders” (Groups 1 and 2) and the “expected poor / suboptimal responders” (Groups 3 and 4) [24,41,42]. Based on a recent systematic review and a further meta-analysis of the data, the addition of rLH could be beneficial for women in POSEIDON Groups 1 and 2, with well preserved ovarian reserve markers and unexpected poor/suboptimal response to rFSH [17]. In our study, almost 30% of patients treated with rLH supplementation belonged to POSEIDON Groups 1 and 2: irrespective of more unfavorable baseline clinical characteristics (older age, lower AMH and AFC) and of a lower oocyte yield, they obtained a cLBR/OPU comparable to women of the same POSEIDON groups receiving rFSH alone. In the present study, about 70% of the women receiving rLH addition belonged to POSEIDON Groups 3 and 4: again, no significant differences were observed in terms of cLBR rate vs. patients of the same groups treated with rFSH alone, despite significantly older age and worse prognosis predictors. Indeed, previous studies suggested rLH supplementation as beneficial specifically in women aged 36–39 years [15,43], and the largest RCT on poor prognosis patients aligned with POSEIDON Group 4 features showed no significant difference in LBR between those stimulated with rFSH plus rLH or rFSH alone [44]; furthermore, the post hoc analysis showed that the poorest ovarian responders had a significantly higher LBR when rLH was added.

## 5. Conclusions

With the limitations of a retrospective analysis, our data suggest that rLH supplementation in COS may represent a reasonable option for patients with predictable or unexpected poor/suboptimal ovarian responsiveness to FSH, those matching the Bologna criteria and those included in the POSEIDON classification. A large, prospective randomized study comparing rFSH alone vs. rFSH plus rLH in this definite category of IVF patients, as well as further basic research studies on the effect of LH on the oocyte and the endometrium, would be welcomed to definitively clarify the proper indication for LH supplementation in COS.

## Figures and Tables

**Figure 1 jcm-11-01575-f001:**
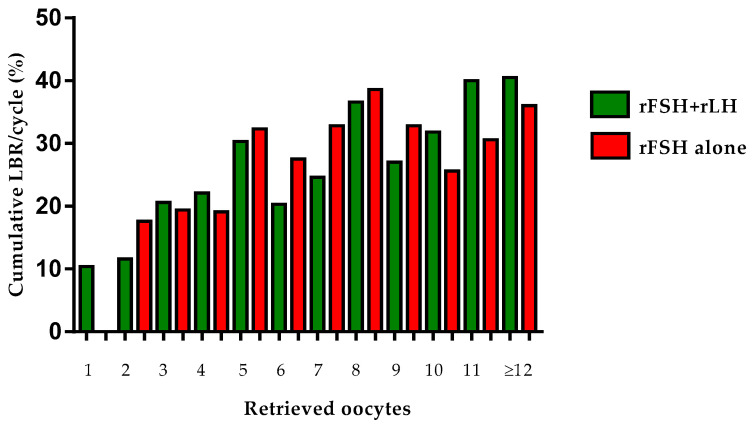
Cumulative live birth rates of all patients with available oocytes (*n* = 1432) receiving COS with rFSH + rLH (*n* = 774) or rFSH alone (*n* = 658), stratified for to the number of retrieved oocytes at OPU.

**Table 1 jcm-11-01575-t001:** Clinical baseline characteristics and IVF outcome of the overall patients’ population (*n* = 1470) receiving COS with rFSH + rLH (*n* = 801) or rFSH alone (*n* = 669). Data are expressed as mean ± standard deviation or as absolute values and percentage, as appropriate.

	rFSH + rLH (*n* = 801)	rFSH Alone (*n* = 669)	*p*
Age (years)	37.8 ± 3.7	35.3 ± 4.2	<0.0001
BMI (kg/m^2^)	22.6 ± 3.6	22.5 ± 3.6	0.22
Basal (day 3) FSH (IU/l)	9.2 ± 3.7	7.5 ± 2.6	<0.0001
AMH (ng/mL)	0.9 ± 0.6	1.6 ± 0.7	<0.0001
Antral follicle count (AFC)	7.9 ± 3.3	11.0 ± 3.3	<0.0001
Previous IVF treatments (*n*)	0.7 ± 1.0	0.3 ± 0.7	<0.0001
Total FSH dose (IU)	1727 ± 602	2321 ± 769	<0.0001
Days of stimulation (*n*)	11.6 ± 2.4	11.4 ± 1.9	<0.05
Peak E2 (pg/mL)	1469 ± 951	1541 ± 1126	0.20
Endometrial thickness (mm)	9.9 ± 2.2	10.1 ± 2.2	0.24
OSI (*n*)	3.4 ± 3.5	3.9 ± 4.0	<0.0001
Retrieved oocytes (*n*)	5.0 ± 3.8	7.9 ± 4.1	<0.0001
Mature (MII) oocytes (*n*)	4.0 ± 2.9	6.3 ±3.3	<0.0001
Maturation rate (%)	81.2 ± 23.2	80.9 ± 19.2	0.73
Fertilization rate (%)	69.4 ± 31.2	68.6 ± 26.6	0.06
Cleavage rate (%)	96.5 ± 14.3	98.1 ± 16.2	0.13
Mean embryo score (*n*)	7.6 ± 1.7	7.5 ± 1.7	0.27
Top quality embryos (%)	52.8 ± 38.7	54.2 ± 34.6	0.62
Frozen embryos (*n*)	0.3 ± 0.9	0.6 ± 1.1	<0.0001
Cycles with no retrieved oocytes % (*n*)	3.4 (27)	1.6 (11)	<0.05
Cycles with no mature oocytes % (*n*)	5.9 (47)	2.2 (15)	<0.0001
Cycles with no fertilized oocytes % (*n*)	14.4 (115)	6.6 (44)	<0.0001
Live birth rate/cycle % (*n*)	19.7 (158)	23.6 (158)	0.07
Cumulative live birth rate/OPU % (*n*)	22.2 (178)	29.3 (196)	<0.01

**Table 2 jcm-11-01575-t002:** Clinical baseline characteristics and IVF outcome of the patients with 6 or less retrieved oocytes (poor or suboptimal responders, *n* = 309) receiving COS with rFSH + rLH (*n* = 263) or rFSH alone (*n* = 46). Data are expressed as mean ± standard deviation or as absolute values and percentage, as appropriate.

	rFSH + rLH (*n* = 263)	rFSH Alone (*n* = 46)	*p*
Age (years)	38.3 ± 3.5	36.4 ± 4.3	<0.01
BMI (kg/m^2^)	22.4 ± 3.0	23.3 ± 4.2	0.43
Basal (day 3) FSH (IU/l)	9.9 ± 4.0	9.0 ± 4.3	0.09
AMH (ng/mL)	0.5 ± 0.3	0.6 ± 0.3	0.25
Antral follicle count (AFC)	4.6 ± 1.4	4.5 ± 1.6	0.77
Previous IVF treatments (*n*)	0.9 ± 1.1	0.7 ± 1.1	0.12
Total FSH dose (IU)	1727 ± 558	2681 ± 716	<0.0001
Days of stimulation	11.4 ± 2.7	11.3 ± 2.3	0.78
Peak E2 (pg/mL)	1230 ± 735	1078 ± 791	0.08
Endometrial thickness (mm)	9.7 ± 2.3	10.1 ± 1.9	0.18
OSI (*n*)	2.3 ± 1.8	1.7 ± 1.3	<0.05
Retrieved oocytes (*n*)	3.8 ± 2.8	4.5 ± 3.0	0.06
Cycles with ≤3 oocytes % (*n*)	55.9 (147)	41.3 (19)	0.07
Cycles with 4–6 oocytes % (*n*)	28.9 (76)	37.0 (17)	0.27
Mature (MII) oocytes (*n*)	3.1 ±2.4	3.5 ± 2.5	0.34
Maturation rate (%)	82.4 ± 25.8	76.2 ± 25.3	0.15
Fertilization rate (%)	68.1 ± 34.5	68.7 ± 29.7	0.79
Cleavage rate (%)	97.8 ± 10.7	95.5 ± 17.3	0.23
Mean embryo score (*n*)	7.6 ± 1.7	6.9 ± 2.3	0.07
Top quality embryos (%)	53.9 ± 39.3	49.0 ± 42.1	0.51
Frozen embryos (*n*)	0.7 ± 1.4	0.1 ± 0.3	0.07
Live birth rate/cycle % (*n*)	14.1 (37)	15.2 (7)	0.84
Cumulative live birth rate/OPU % (*n*)	15.6 (41)	15.2 (7)	0.95

**Table 3 jcm-11-01575-t003:** Cumulative live birth rates of all patients with available oocytes (*n* = 1432) receiving COS with rFSH + rLH (*n* = 774) or rFSH alone (*n* = 658), stratified for to the number of retrieved oocytes at OPU. Data are expressed as absolute values and percentage.

Cumulative LBR/OPU % (*n*)			
Retrieved	rFSH + rLH	rFSH Alone	
Oocytes	(*n* = 774)	(*n* = 658)	*p*
1	10.4 (7/67)	0 (0/11)	0.26
2	11.6 (13/112)	17.6 (6/34)	0.36
3	20.6 (20/97)	19.4 (7/36)	0.88
4	22.1 (29/131)	19.1 (9/47)	0.67
5	30.3 (23/76)	32.3 (20/62)	0.80
6	20.3 (13/64)	27.5 (19/69)	0.33
7	24.6 (16/65)	32.8 (21/64)	0.30
8	36.6 (15/41)	38.6 (27/70)	0.84
9	27.0 (10/37)	32.8 (20/61)	0.55
10	31.8 (7/22)	25.6 (11/43)	0.59
11	40.0 (8/20)	30.6 (11/36)	0.47
≥12	40.5 (17/42)	36.0 (45/125)	0.60

**Table 4 jcm-11-01575-t004:** Clinical baseline characteristics and IVF outcome of the overall patients’ population (*n* = 1470) receiving COS with either rFSH + rLH (*n* = 801) or rFSH alone (*n* = 669), stratified according to the POSEIDON classification. Data are expressed as mean ± standard deviation or as absolute values and percentage, as appropriate.

	rFSH + rLH	rFSH Alone	
	(*n* = 801)	(*n* = 669)	*p*
**POSEIDON Group 1**	**(*n* = 55, 7%)**	**(*n* = 212, 32%)**	
Age (years)	31.0 ± 2.5	31.3 ± 2.3	0.46
AMH (ng/mL)	1.9 ± 0.5	1.9 ± 0.4	0.38
Antral follicle count (AFC)	10.9 ± 3.2	12.1 ± 2.9	<0.05
Retrieved oocytes (*n*)	7.8 ± 4.5	9.2 ± 4.3	<0.05
Mature (MII) oocytes (*n*)	6.1 ± 4.2	7.2 ± 3.3	0.08
Fertilization rate (%)	71.2 ± 28.8	70.8 ± 23.5	0.92
Top quality embryos (%)	47.6 ± 34.0	56.2 ± 33.8	0.11
Cumulative live birth rate/OPU % (*n*)	36.4 (20/55)	38.2 (81/212)	0.80
**POSEIDON Group 2**	**(*n* = 160, 20%)**	**(*n* = 269, 40%)**	
Age (years)	38.5 ± 2.3	38.0 ± 2.4	<0.05
AMH (ng/mL)	1.8 ± 0.4	1.9 ± 0.4	<0.01
Antral follicle count (AFC)	9.9 ± 3.2	11.1 ± 2.9	<0.001
Retrieved oocytes (*n*)	7.0 ± 3.7	8.2 ± 3.8	<0.01
Mature (MII) oocytes (*n*)	5.5 ± 3.0	6.5 ± 3.2	<0.01
Fertilization rate (%)	68.8 ± 25.7	69.4 ± 26.6	0.84
Top quality embryos (%)	52.1 ± 36.5	52.4 ± 33.1	0.94
Cumulative live birth rate/OPU % (*n*)	24.4 (39/160)	26.0 (70/269)	0.70
**POSEIDON Group 3**	**(*n* = 81, 10%)**	**(*n* = 61, 9%)**	
Age (years)	31.8 ± 1.9	31.2 ± 3.2	0.22
AMH (ng/mL)	0.6 ± 0.3	0.8 ± 0.2	<0.0001
Antral follicle count (AFC)	7.1 ± 3.2	10.2 ± 3.6	<0.001
Retrieved oocytes (*n*)	4.9 ± 3.0	6.8 ± 3.4	<0.01
Mature (MII) oocytes (*n*)	3.7 ± 2.2	5.5 ± 3.0	<0.001
Fertilization rate (%)	63.7 ± 30.7	62.6 ± 30.7	0.83
Top quality embryos (%)	60.7 ± 39.6	57.0 ± 39.4	0.60
Cumulative live birth rate/OPU % (*n*)	29.6 (24/81)	26.2 (16/61)	0.66
**POSEIDON Group 4**	**(*n* = 505, 63%)**	**(*n* = 127, 19%)**	
Age (years)	39.2 ± 2.4	38.4 ± 2.7	<0.01
AMH (ng/mL)	0.6 ± 0.3	0.7 ± 0.3	<0.0001
Antral follicle count (AFC)	7.0 ± 2.9	8.6 ± 3.8	<0.001
Retrieved oocytes (*n*)	4.1 ± 2.9	5.3 ± 3.2	<0.0001
Mature (MII) oocytes (*n*)	3.4 ± 2.5	4.5 ± 3.0	<0.001
Fertilization rate (%)	70.2 ± 33.1	65.5 ± 29.0	0.13
Top quality embryos (%)	52.4 ± 39.8	53.0 ± 36.6	0.88
Cumulative live birth rate/OPU % (*n*)	18.8 (95/505)	22.8 (29/127)	0.31

## Data Availability

The datasets used and/or analyzed during the current study are available from the corresponding author on reasonable request.

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
