# Peer review of "Effect of rLH Supplementation during Controlled Ovarian Stimulation for IVF: Evidence from a Retrospective Analysis of 1470 Poor/Suboptimal/Normal Responders Receiving Either rFSH plus rLH or rFSH Alone"

_jcm, 2022, doi:10.3390/jcm11061575_

Round 1

Reviewer 1 Report

Research into the role of LH in controlled ovulation stimulation has a long history. Despite numerous studies, there is no clear conclusion. FSH is commonly used for ovarian stimulation; however, the role of supplementary LH is still controversial. LH plays a key role in the intermediate-late phases of folliculogenesis et maybe on quality of oocyte . Although ovarian stimulation is efficiently achieved in most cases by the administration of exogenous FSH alone. Usually, in elderly patients and with a lower ovarian reserve, most preparations containing LH or low doses of hCG are added. According the results, of this work authors  indicate groups of POSEJDON  classification  that can benefit from the LH supplement In my opinion The papier is one of the numerous work on the topic with little impact on clinicians' beliefs and with weak evidence when supplementing with LH

Author Response

For sure the role of LH in COS is not yet well understood, despite several original research papers and some meta-analyses have been already published. It seems rather clear, however, that LH administration is not beneficial for all patients undergoing IVF, and therefore, the effort now should be to categorize those patients who will effectively benefit from LH administration. Our work goes in this direction. We think that the results reported herein wil not solve the issue, nor will be resolutive of all controversies, and maybe will not convince scepticals; anyway, they still are worth publication because they clearly show how LH adjunct is useful for poor prognosis patients of specific Poseidon groups, whose final results in terms of live births (the outcome that most matters) become comparable to that of patients with better prognostic indexes. Moreover, to the best of our knowledge, there are no numerous works comparing rFSH vs. rFSH+rLH stratifying IVF patients according to Poseidon classification in a single setting, where all were treated by the same Lab and the same physicians.  

Reviewer 2 Report

In this study the authors investigated the effectiveness of administering FSH+LH in comparison to FSH alone in normal, suboptimal and poor responders.

Despite the study is not really novel, the results presented in this manuscript are important to have alternative treatments, especially in poor responder women. The manuscript is well written, easy to follow and the results are clearly presented. Despite these findings higlight the greatest effectivennes of FSH treatment in normal population, the co-administration of FSH- LH may preserve oocyte quality in poor responder women and hence LH seems to play a crucial role to obtain equal cumulative live birth rates. 

Author Response

Thanks for your comments.

Reviewer 3 Report

The authors present a retrospective study of women treated with IVF in order to compare live birth rate between women whose ovarian stimulation included rLH and those that did not.  This paper is well written and clear in describing the methods.  They also do not overreach in their conclusion which is appropriate given the retrospective nature of the study.  I like the primary outcome that was chosen because it is the outcome of most interest to patients.  I don't know if the analysis of only suboptimal and poor responders was planned at the outset of the study, but I do think sufficient justification for looking at the data in this way was given.

Here are some comments:

With only normal BMI included, your study population is not very generalizable to my study population. Could you please discuss how many total cycles were available for review within the study period, and then based on your inclusion criteria how many were excluded and for what reasons?

With a retrospective study, the patients will by necessity typically have undergone treatment in a "real-life" setting, as part of routine clinical practice. I do not believe you need to specify or use this term within the title or the body of the report.

You describe the protocols used in the methods, but in the results I do not see a clear report of how many patients underwent an agonist versus an antagonist protocol. I cannot tell whether one protocol was more frequently chosen for normal or poor responders, or whether choice of protocol had any effect on the success of cycles with rLH. It would be helpful to at least include this in the baseline characteristics, and ideally to take this into account in some way in your comparison. If the authors do not believe this would have contributed, please explain.

Author Response

Local rules regulating IVF in our administrative area do not allow to perform IVF in publis settings (almost no charge for patients) when the BMI is below 18 or above 30. We further restricted the inclusion criteria to BMI 18-25 in oder to avoid any effect of "dilution" of gonadotropins that could act as confounder of the real effectiveness of the medication. Among a total number of 1612 women undergoing IVF in the time period of the study, 242 were excluded because not matching the inclusion criteria. We added this statement in the text showing also the reasons for exclusion (see Study design)   We agree that it is not necessary to stress the real life nature of the study, and we eliminated it from the title and the text, just saying "retrospective".   We added in the text the relative proportion of cycles performed with long protocol or with GnRH-antagonist protocol. In none of our previous studies, and in this one also, we observed a siginificantly different effectiveness of IVF according to the type of pituitary suppression; therefore, we did not indicate which kind of pituitary block was applied in each patients' subgroup, but the proportion was the same as in the general patients' population. We added a statement in Methods.